*Report*

EMBO
Molecular Medicine

# FGF19 and its analog Aldafermin cooperate with MYC to induce aggressive hepatocarcinogenesis

José Ursic-Bedoya [ID][1,2,5], Guillaume Desandré[1,5], Carine Chavey[1], Pauline Marie [ID][1], Arnaud Polizzi[3], Benjamin Rivière[4], Hervé Guillou [ID][3], Eric Assenat[1,2], Urszula Hibner [ID][1] & Damien Gregoire [ID][1✉]

## Abstract

**FGF19 hormone has pleiotropic metabolic functions, including the modulation of insulin sensitivity, glucose/lipid metabolism and energy homeostasis. On top of its physiological metabolic role, FGF19 has been identified as a potentially targetable oncogenic driver, notably in hepatocellular carcinoma (HCC). Nevertheless, FGF19 remained an attractive candidate for treatment of metabolic disease, prompting the development of analogs uncoupling its metabolic and tumor-promoting activities. Using pre-clinical mice models of somatic mutation driven HCC, we assessed the oncogenicity of FGF19 in combination with frequent HCC tumorigenic alterations: p53 inactivation, CTNNB1 mutation, CCND1 or MYC overexpression. Our data revealed a strong oncogenic cooperation between FGF19 and MYC. Most importantly, we show that this oncogenic synergy is conserved with a FGF19-analog Aldafermin (NGM282), designed to solely mimic the hormone's metabolic functions. In particular, even a short systemic treatment with recombinant proteins triggered rapid appearance of proliferative foci of MYC-expressing hepatocytes. The fact that FGF19 analog Aldafermin is not fully devoid of the hormone's oncogenic properties raises concerns in the context of its potential use for patients with damaged, mutation-prone liver.**

**Keywords** Aldafermin (NGM282); FGF19-15; Hydrodynamic Injections; Liver Cancer; Oncogenic Cooperation
**Subject Categories** Cancer; Digestive System; Metabolism

## Introduction

Fibroblast growth factor 19 (FGF19) is an ileum-secreted hormone that controls bile acids synthesis and regulates several metabolic functions (Owen et al, 2015). Pleiotropic activities of FGF19, and its rodent ortholog FGF15, are important regulators of liver physiology, through increase of metabolic rate, improvement of glucose and

insulin tolerance and decrease of fasting insulin levels (Lan et al, 2017; Marcelin et al, 2014; Gadaleta and Moschetta, 2019). On the other hand, FGF19 displays a mitogenic activity and has been identified as an oncogenic driver in hepatocellular carcinoma (HCC). Focal amplification of 11q13 locus, encompassing *FGF19*, is among the most frequent amplification events in HCC (6–15% of HCC) (Sawey et al, 2011; Schulze et al, 2015; Guichard et al, 2012). This genetic alteration is associated with more aggressive tumors, higher risk of recurrence after surgical resection and lower overall survival rates (Miura et al, 2012; Ahn et al, 2014; Schulze et al, 2015; Kang et al, 2019). Moreover, mice with forced FGF19 overexpression develop hepatic tumors (Nicholes et al, 2002; Zhou et al, 2014a, 2017c) while targeting of FGFR4, the main FGF19 receptor in the liver, reduces tumor growth (Hagel et al, 2015; French et al, 2012). Importantly, an inhibitor of FGFR4, Fisogatinib (BLU-554), has demonstrated clinical activity in HCC patients with aberrant FGF19 expression (Kim et al, 2019) (phase I clinical trial, NCT02508467).

The interconnections between FGF19 oncogenic and metabolic activities, as well as the relative contributions of the downstream effectors of the hormone, remain poorly characterized. FGF19 binding to its receptor FGFR4 and co-receptor β-Klotho (KLB) has been reported to activate several major signal transduction pathways, such as Ras/MAPK, Akt, Mst1/2 and β-catenin signaling, which are all implicated in hepatic carcinogenesis (Chen et al, 2020; Ornitz and Itoh, 2015; Ji et al, 2019). Moreover, FGF19 acts on multiple cell types and binds other members of the FGFR family, including FGFR1c (Owen et al, 2015; Lan et al, 2017). This redundancy between FGFRs may account for some limitations in the efficacy of FGFR4 specific inhibitors (Tao et al, 2022).

Despite the clear association between FGF19 and HCC, the metabolic properties of the hormone, notably its remarkable inhibitory effect on lipid accumulation, have prompted efforts to develop analogs devoid of the oncogenic activity. (Gadaleta et al, 2018; Zhou et al, 2014b). NGM282 (more recently named Aldafermin), has been reported to retain bile acid synthesis inhibition, without promoting tumor formation (Zhou et al, 2014b, 2017a; DePaoli et al, 2019). Aldafermin, which was selected in a systematic screen of FGF19 mutants in diabetic db/db mice, carries a 5-amino acid deletion and 3 amino acids substitutions in the N-terminal part of the protein, which is the FGFR4 receptor

[1]Institut de Génétique Moléculaire de Montpellier, University of Montpellier, CNRS, Montpellier, France. [2]Department of Hepatogastroenterology, Hepatology and Liver Transplantation Unit, Saint Eloi Hospital, CHU Montpellier, University of Montpellier, Montpellier, France. [3]Toxalim (Research Center in Food Toxicology), INRAE, ENVT, INP-PURPAN, UMR 1331, UPS, Université de Toulouse, Toulouse, France. [4]Department of Pathology, Gui de Chauliac Hospital, CHU Montpellier, University of Montpellier, Montpellier, France. [5]These authors contributed equally: José Ursic-Bedoya, Guillaume Desandré. ✉E-mail: damien.gregoire@igmm.cnrs.fr

binding domain (Zhou et al, 2014b). The proposed interpretation for the reported loss of the oncogenic activity of Aldafermin was that the altered binding to the FGFR4/KLB receptor would fail to activate the STAT3 pathway, which might be implicated in the oncogenic properties of FGF19 (Zhou et al, 2017c, 2014b). Aldafermin was subsequently tested in several clinical trials involving patients receiving the molecule subcutaneously during 12 or 24 weeks for non-cirrhotic metabolic dysfunction-associated steatohepatitis (MASH), primary sclerosing cholangitis, primary biliary cholangitis or bile acid diarrhea (Hirschfield et al, 2019; Harrison et al, 2022; Mayo et al, 2018; BouSaba et al, 2023). These first trials generated promising results with decrease in liver fat content and fibrosis serum surrogate markers. A phase 2b randomized clinical trial involving 160 patients with MASH-related compensated cirrhosis treated with Aldafermin for 48 weeks has just been published (Rinella et al, 2023). While the primary end point has been achieved (decrease in enhanced liver fibrosis score ELF in the 3 mg Aldafermin-treated group), it failed to show histological improvement.

Here, we investigated the effects of FGF19/15 and Aldafermin in the framework of oncogenic cooperation with several common HCC drivers. We reasoned that a diseased, inflamed liver is subjected to microenvironmental insults that may result in increased somatic mutation incidence, as shown in the context of cirrhosis (Rebouissou and Nault, 2020). It was therefore of interest to assess possible tumor promoting effects of oncogenic combinations involving FGF19 and its analogs.

# Results

## FGF19 and FGF15 cooperate with MYC to trigger hepatic carcinogenesis

We used hydrodynamic gene transfer (HGT) (Zhang et al, 1999; Liu et al, 1999) to transfect hepatocytes in vivo to combine overexpression of the human FGF19 with genetic alterations frequently observed in HCC. We selected Trp53 inactivation, β-catenin activating mutation and MYC overexpression, which are among the most frequently affected oncogenic alterations in human HCC (Zucman-Rossi et al, 2015; Molina-Sánchez et al, 2020; Fig. 1A), as well as the overexpression of cyclin D1 (CCND1), the latter mimicking the focal amplification of 11q13 (encoding both FGF19 and CCND1) observed in 10% of HCC (Sawey et al, 2011).

FGF19 transfection of hepatocytes led to supra-physiological plasma levels of the hormone (1–10 ng/mL), that persisted over several months and displayed long-term metabolic effects (Ursic-Bedoya et al, 2022). Nine-to-12 months after HGT, most animals developed tumors (incidence $n = 5/6$, one tumor/liver) (Fig. 1B). Thus, with circulating FGF19 levels 6 to 100-fold lower than in previously described mouse models of FGF19 overexpression (Zhou et al, 2017b; Nicholes et al, 2002), we confirmed the previously reported moderate oncogenic effect of FGF19.

Associating FGF19 overexpression with Crispr/Cas9-mediated inactivation of p53 (Bacevic et al, 2019), did not accelerate tumorigenesis (Fig. 1C). Similarly, the incidence of tumor development in FGF19/CCND1/sgTrp53 animals 10 months after HGT remained identical to that of FGF19 animals ($n = 2/5$,

maximum of 1 tumor/liver) (Fig. 1C). Therefore, our results suggest that there is no oncogenic cooperation between FGF19 overexpression and p53 inactivation or CCND1 overexpression.

In contrast, combination of overexpression of FGF19 and β-catenin$^{S33Y}$ gave rise to tumors in 4 months ($n = 9/9$, mean = 3.7 tumors/liver, Figs. 1C and EV1), significantly increasing hepatic carcinogenesis that was triggered by β-catenin$^{S33Y}$ alone at the same time point ($n = 2/6$, 1 or 2 tumors/liver).

However, the most spectacular results were obtained when FGF19 was associated with the overexpression of MYC: at a maximum of 4 weeks after HGT, these animals had to be sacrificed, for ethical reasons (Fig. 1C). Their livers presented an elevated tumor burden ($n = 11/12$; mean = 72 tumors per liver, CI$_{95}$ [45, 99]) (Fig. 1D,E). This was in sharp contrast with control mice transfected with MYC and RFP (i.e., control empty vector) for which only 2 mice out of 10 presented tumors (1 tumor per liver), despite the extended delay of their sacrifice, up to 14 weeks post-HGT (Fig. 1D,E).

Oncogenic properties of FGF15, the rodent ortholog of FGF19, are controversial. While one report found that FGF15 enhanced chemically induced liver carcinogenesis (Uriarte et al, 2015), another article stated that FGF15 was devoid of oncogenic properties, presumably due to structural differences with the human hormone (Zhou et al, 2017b). In our model, the co-expression of FGF15 with MYC also gave rise to hepatic carcinogenesis, albeit within a slightly longer timeframe (7/8 mice exhibited tumors after 4–6 weeks post-HGT) and a lower tumor burden (mean = 10.3) in comparison with FGF19 + MYC (Fig. 1D,E). Anatomopathological and biochemical analyses revealed that the FGF19 + MYC and FGF15 + MYC tumors were both moderately differentiated and expressed HCC markers Alpha-fetoprotein (Afp) and Glypican 3 (Gpc3) (Fig. 1F). We conclude that FGF19 and FGF15 cooperate with MYC to trigger rapid and aggressive hepatic carcinogenesis.

## Aldafermin is oncogenic when combined with MYC overexpression

Our discovery of significant oncogenic activity of FGF15 prompted us to have a closer look at the FGF19 analog Aldafermin (NGM282). Of note, the sequence homology between FGF19 and Aldafermin is high, considerably higher than between the human and mouse orthologs. Indeed, there are only 8 AA differences between the natural hormone and its pharmacological analog and the AlphaFold predicted 3D structures of the two proteins display 97% similarity (Fig. 2A; Jumper et al, 2021).

We therefore tested the oncogenicity of Aldafermin hepatic expression, either alone or in combination with MYC (Fig. 2B). As expected, Aldafermin overexpression alone did not trigger tumor development after 2–4 weeks ($n = 0/12$). In striking contrast, the majority of livers co-transfected with Aldafermin and MYC exhibited multiple tumors within this short timeframe ($n = 10/13$; tumors mean number = 7.9; CI$_{95}$ [1.3, 14.5]). HES staining revealed that tumors were moderately differentiated and were in fact indistinguishable from the FGF15/19 tumors by an anatomopathological examination (Fig. 2B). To further examine this point, we performed gene expression analyses of the tumors by bulk-RNAseq ($n = 5$ for each). Principal Component Analysis of gene expression profiles and correlation analysis indicate that Aldafermin + MYC

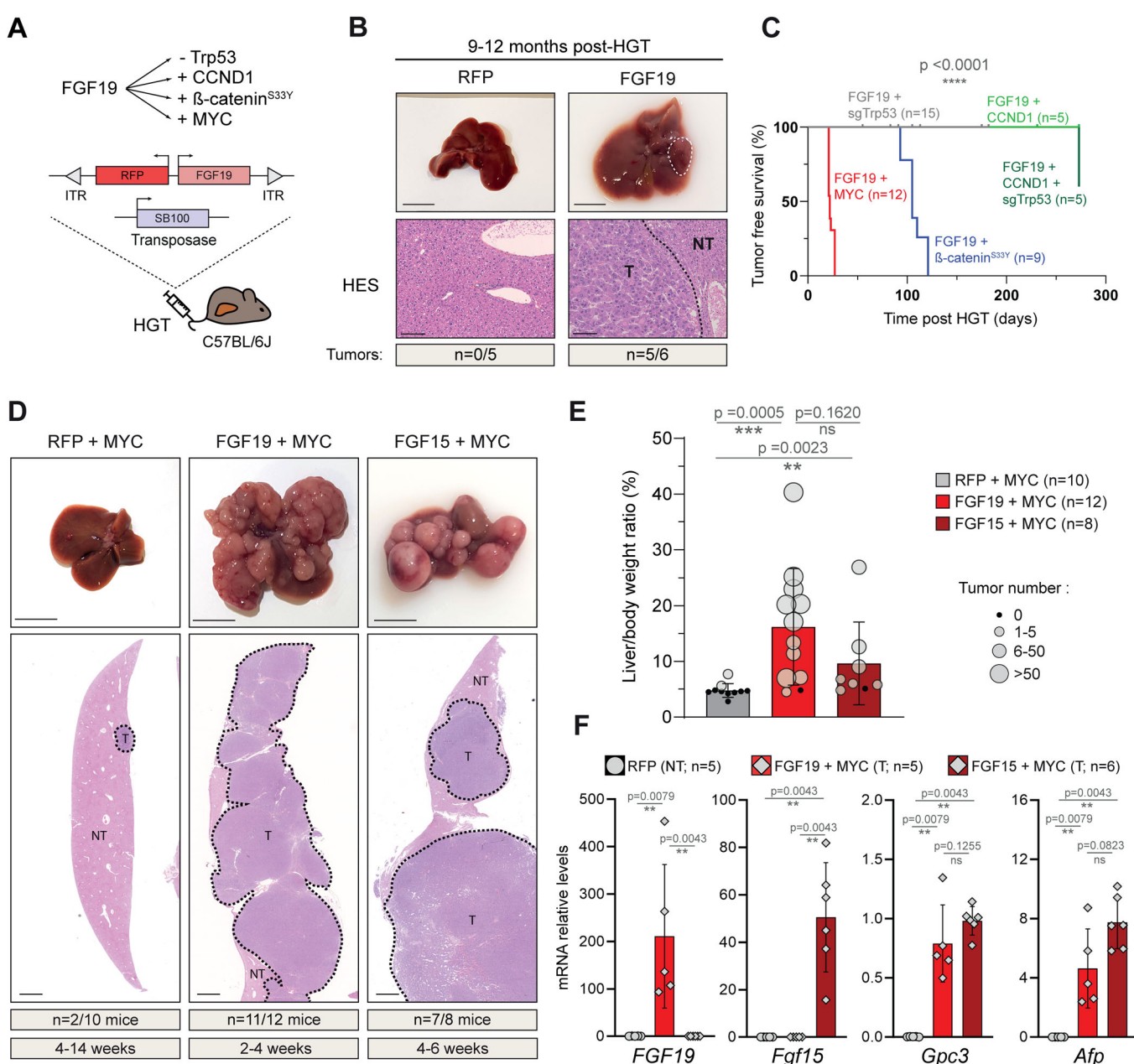

**Figure 1. FGF19 and FGF15 cooperate with MYC to trigger liver carcinogenesis.**

(A) Experimental strategy to test FGF19/15 cooperation with other oncogenic events. Female C57BL/6J mice are subjected to hydrodynamic gene transfer (HGT) of a combination of oncogenes and the Sleeping-Beauty transposase 100 (SB100). (B) Representative livers and hematoxylin & eosin & Saffron (HES) stained sections following HGT with empty vector (RFP) or FGF19 plasmids, as indicated. (C) Kaplan–Meier curve of tumor-free survival for distinct oncogenic combinations, as indicated. Statistical significance was determined using Mantel-Cox (Log-rank) test; $p$ value is indicated. (D) Representative livers and HES stained sections following hydrodynamic injection with RFP + MYC, FGF19 + MYC and FGF15 + MYC. (E) Liver/body weight ratio and tumor number per liver following HGT with the indicated combinations of oncogenes. (F) RT-qPCR expression analysis of transfected FGF 19/15 and HCC markers Glypican 3 and Alpha fetoprotein in RFP parenchyma (NT) and FGF19 + MYC and FGF15 + MYC tumors (T). Data information: Scale-bars: 1 cm (macroscopic liver), 100 μm (HES staining). Data are represented as mean ± SD. Number of mice per group and tumor incidence indicated are indicated in the figure. The data are representative of at least two independent experiments and all replicates shown are biological replicates. All tumoral samples come from independent mice. Statistical significance was evaluated using Mann–Whitney test performed on liver/body weight (E) or mRNA levels (F). Significant $p$ values are indicated: ns (non significant) $p > 0.05$, **$p < 0.01$, ***$p < 0.001$, ****$p < 0.0001$. RFP red fluorescent protein, T tumor, NT non-tumor parenchyma. Source data are available online for this figure.

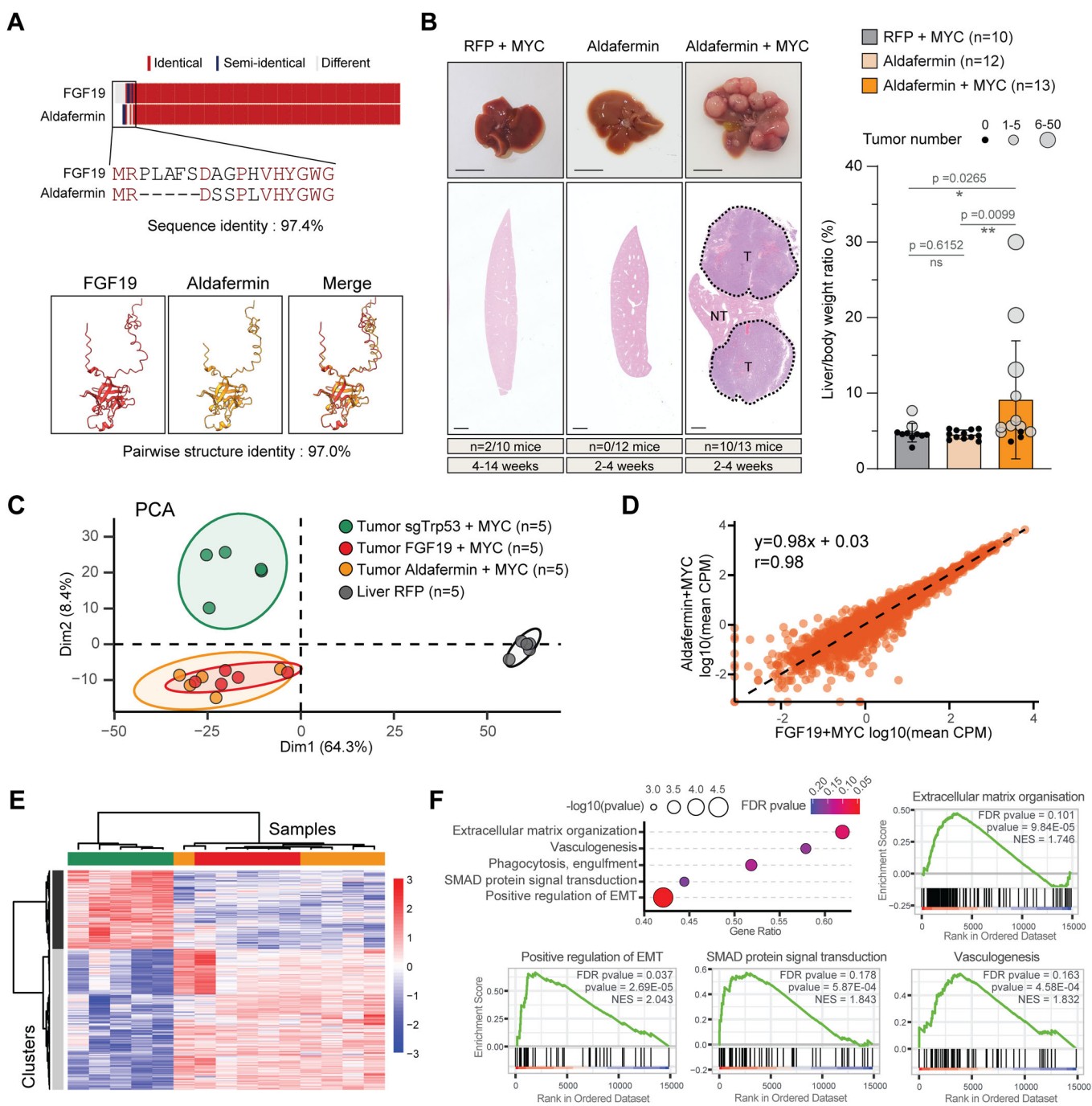

**Figure 2. Aldafermin and FGF19 are indistinguishable in cooperating with MYC to promote liver carcinogenesis.**

(A) Sequence alignment of FGF19 and Aldafermin proteins. Amino-acid sequences of N-terminal part of the proteins are indicated. 3D AlphaFold prediction of proteins' structure and folding are shown. (B) Representative livers, HES stained sections, liver/body weight ratio and tumor number per liver following HGT with MYC + RFP, Aldafermin or Aldafermin + MYC, 2–4 weeks after injection. Of note, quantifications for MYC + RFP tumors were already presented in Fig. 1E. (C) Principal Component Analysis of RNAseq gene expression profiles of tumors obtained after HGT with FGF19 + MYC, Aldafermin + MYC, sg-Trp53 + MYC or liver (HGT with RFP plasmid) as control. (D) Correlation plot of the mean log10(CPM) of the Aldafermin + MYC and FGF19 + MYC tumors. Each point represents one gene, and the black line is a linear regression over the whole dataset. Equation of the linear regression and its Pearson's correlation coefficient are indicated. (E) Heat map showing hierarchical clustering of genes ($n = 786$) separating tumoral transcriptomes along dimension 2 of the PCA (contribution to Dim.2 > 0.003 and |correlation to Dim.2| > 0.5). (F) Gene set enrichment analysis (GSEA) pathways specifically deregulated in Aldafermin/FGF19 driven tumors. A selection of most significant pathways is shown. Data information: Data are presented as mean ± SD. Mann–Whitney test statistical significance is indicated. ns (non significant) $p > 0.05$, *$p < 0.05$, **$p < 0.01$. Scale bars: 1 cm. All tumoral samples for RNA-seq come from independent mice, all replicates shown are biological replicates. Data from at least two independent experiments. HES Hematoxylin & eosin & saffron. GSEA analysis were performed with ViSEAGO package. Source data are available online for this figure.

and FGF19 + MYC tumors have indistinguishable transcriptomic profiles (Fig. 2C,D). To gain insight into the pro-carcinogenic effects of the hormone and its analog, we next wished to compare gene expression profiles of FGF19 or Aldafermin-driven tumors with those triggered by other oncogenic combinations involving MYC. Since MYC overexpression alone only rarely gave rise to tumors, we chose to use the combination of MYC expression with CRISPR-mediated inactivation of the p53 tumor suppressor (Fig. 2C). We identified 786 genes separating MYC + SgTrp53 from MYC + FGF19 or MYC + Aldafermin tumors (contribution to Dim.2 > 0.003 and |correlation to Dim.2| > 0.5) (Fig. 2E). By comparing with liver transcriptomes, we minimized the contribution of p53 deficiency and looked for pathways specifically deregulated in FGF19/Aldafermin-driven tumors (Fig. EV2A). We identified deregulation of 18 GSEA pathways ($p$ value <0.01) (Figs. 2F and EV2B, Dataset EV1) among which positive regulation of epithelial-to-mesenchymal transition, vasculogenesis and extracellular matrix remodeling (Fig. 2D) are particularly interesting and warrant further investigations.

Overall, our results reveal that the modification of 8 AA at the N-terminal part of FGF19, producing the FGF19 analog Aldafermin, does not abolish oncogenic properties of the hormone, at least in the context of co-expression with MYC.

### Oncogenic effects of systemic administration of FGF19 and Aldafermin recombinant proteins

While FGF19 is expressed by hepatocytes in a significant proportion of HCC, hepatic expression of a transgene encoding Aldafermin is clearly not physiological. We therefore designed an experimental scheme to test the mitogenic effects of Aldafermin and FGF19 in a setup more closely resembling the clinical reality, where patients receive Aldafermin daily (Harrison et al, 2020; Mayo et al, 2018; Hirschfield et al, 2019; Harrison et al, 2018).

We produced tag-free recombinant FGF19 and Aldafermin (Choi et al, 2023, 2020; Fig. 3A) and performed intra-peritoneal injections of 2 µg of purified proteins per mouse (corresponding to the 3 mg of the drug given to patients in clinical trials (Harrison et al, 2018)). Transcriptional repression of hepatic $Cyp7a1$, a bona fide target of FGF19, confirmed the biological activity of the recombinant proteins in vivo (Fig. 3B). A single injection led to circulating levels of around 20 ng/mL after 2 h, in accordance with the reported short half-life of the hormone (Degirolamo et al, 2016). To approach plasma levels observed in Aldafermin-treated patients, we next injected 1.2 µg of recombinant proteins every 12 h. We tested the effect of FGF19 or Aldafermin administration on animals previously subjected to hepatocyte transfection by HGT with MYC plasmid (Fig. 3C). After 4 days, livers of mice from the control group (HGT-MYC followed by rGST injection) had no detectable tumoral foci and very few MYC-positive cells, which was expected, since these cells are rapidly eliminated in the absence of a cooperating oncogenic event. In contrast, mice that received either FGF19 or Aldafermin presented numerous MYC-positive foci (mean size = 0.026 mm$^2$; CI$_{95}$ [0.015, 0.037] and 0.026 mm$^2$; CI$_{95}$ [0.020, 0.031] for FGF19 and Aldafermin respectively) (Fig. 3C). EdU incorporation indicated that these foci were highly proliferative, and that Aldafermin effects were indistinguishable from FGF19 (Fig. 3C). Thus, Aldafermin and FGF19 show similar

pro-tumorigenic activities on MYC expressing hepatocytes, confirming that the metabolic and oncogenic activities of the analog of the hormone have not been fully uncoupled.

## Discussion

In this study, we have identified a strong oncogenic cooperation between the FGF19 hormone and MYC, driving hepatocarcinogenesis. Importantly, we discovered that FGF19 analog Aldafermin conserves oncogenic properties in this context.

Among all the combinations tested, FGF19 and MYC oncogenic cooperation stood out, as it triggered an impressively fast and aggressive tumorigenesis. As FGF19 tumor-promoting effects were observed in the context of cooperation with MYC or β-catenin, both of which have a documented pro-mitogenic activity, our result suggests that they are unlikely to be solely based on stimulation of cell proliferation. The precise molecular mechanisms responsible for these synergistic effects remain to be elucidated. In addition, analysis of tumor transcriptomes suggests that induction of EMT, SMAD protein signal transduction, reorganization of extracellular matrix and vasculogenesis are enriched in FGF19/Aldafermin + MYC driven tumors. Further investigations will be required to decipher the contributions of these potential deregulations to pro-carcinogenic effects of the hormone and its analog. Of note, while aberrant FGF19 expression is frequent in HCC, other types of tumors, such as head and neck squamous cell carcinoma, have been shown to be driven by FGF19 (Gao et al, 2019), our findings might therefore also be of interest in these contexts.

Aldafermin is not the only FGF19 analog described as non-mitogenic. Other variants of the hormone that have been engineered (Wu et al, 2010; Gadaleta et al, 2018; DePaoli et al, 2019) all carry changes in the N-terminal part of the protein, as these modifications have been described to curtail the oncogenic activity of FGF19 due to a weakened activation (dimerization) of the FGFR4 receptor (Jin et al, 2023). However, our data indicate that, in the context of MYC overexpression, the signaling by Aldafermin to its downstream effectors is sufficient to promote oncogenic growth. Of note, considering the functional redundancy between FGF19 and FGF21, and their shared specificity for the FGFR receptors (FGFR4, FGFR1c), it may be of interest to further investigate possible oncogenic cooperation of FGF21 analogs.

Finally, it is important to consider the relevance of oncogenic cooperation in a clinical setting. Aldafermin, as well as other FGF19 analogs, are designed to treat patients with damaged liver. There is evidence for clonal expansion in MASH livers (Wang et al, 2023) and cirrhosis is associated with accumulation of mutations (Schulze et al, 2015). MYC target gene signature has recently been reported to be associated with non-cirrhotic MASH-induced HCC (Pinyol et al, 2021). Altogether, MYC amplification is found in 6–17% of human HCC (Ally et al, 2017), while MYC pathway is activated in close to 30% of human HCC (Kaposi-Novak et al, 2009), making it a relevant alteration in the context of oncogenic cooperation.

We believe that our results showing that Aldafermin retains oncogenic properties raise significant concerns. They indicate that the benefits of treating metabolic liver disorders with FGF19 analogs need to be carefully balanced against the drugs' possible direct effects on carcinogenesis.

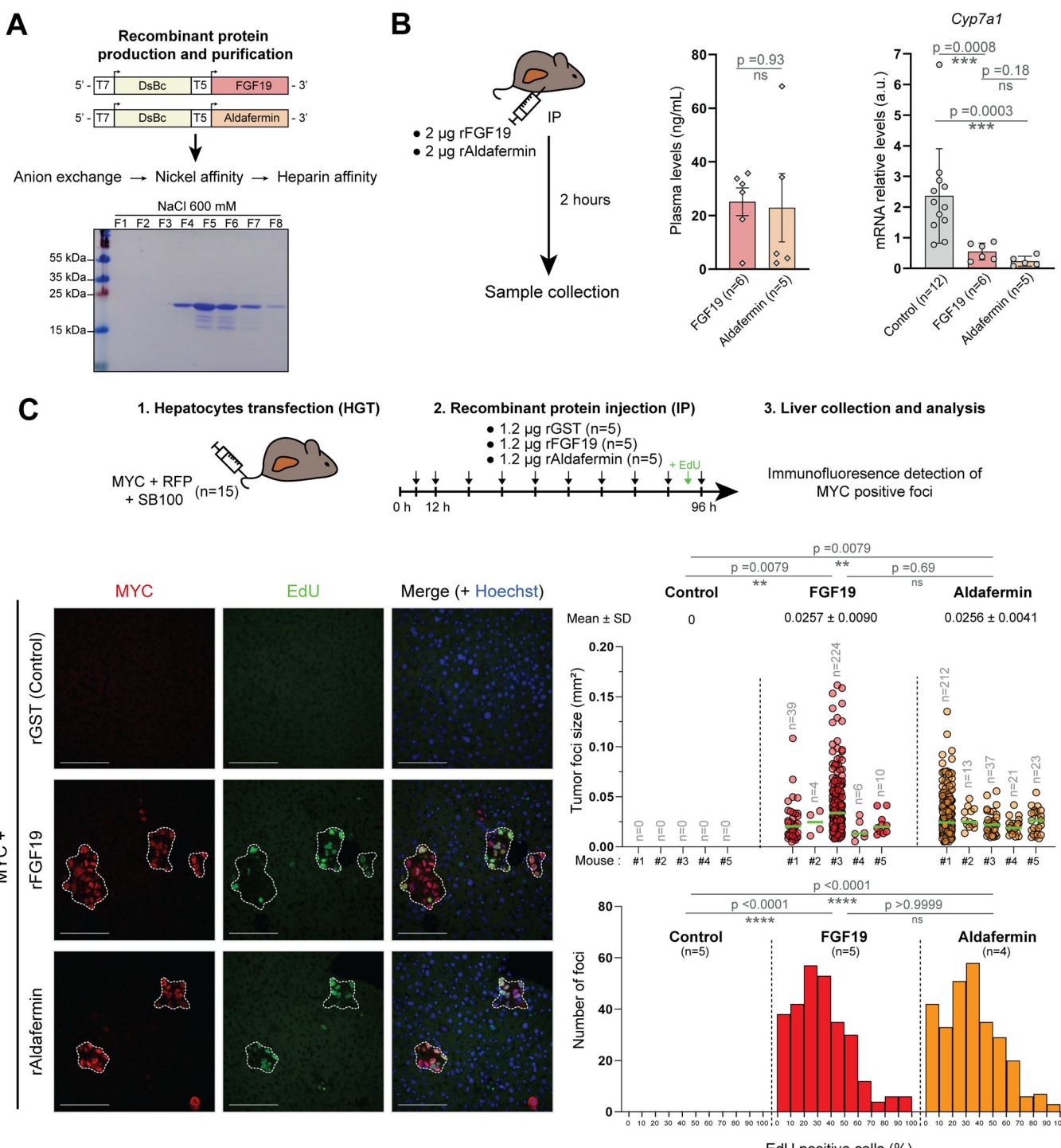

## Methods

### Mice experiments

All reported animal procedures were carried out in accordance with the rules of the French Institutional Animal Care and Use Committee and European Community Council (2010/63/EU). Animal studies were approved by institutional ethical committee (Comité d'éthique en

expérimentation animale Languedoc-Roussillon (#36)) and by the Ministère de l'Enseignement Supérieur, de la Recherche et de l'Innovation (D. Gregoire: APAFIS #32384-2021070917596346 v2). ARRIVE guidelines were followed. Mice were housed in groups of 5 maximum per cage, and were provided ad libitum food (standard diet) and water access. Rooms for housing were temperature-controlled (22 °C) with a 12 h light/dark cycle. Mice were fasted 5 h before euthanasia and liver collection.

**Figure 3. Systemic administration of recombinant FGF19 and Aldafermin recapitulates cooperation with MYC.**

(A) Schematic representation of the workflow for recombinant FGF19 and Aldafermin production and purification. (B) ELISA quantification of plasma levels of FGF19 and Aldafermin 2 h after intraperitoneal injection of 2 μg of recombinant FGF19 or Aldafermin. RT-qPCR quantification of liver *Cyp7a1* mRNA levels 2 h post injection. (C) Schematic representation of the experiment. After hydrodynamic gene transfer of MYC, RFP and SB100X encoding plasmids, mice were intraperitoneally injected every 12 h with either 1.2 μg rFGF19, rAldafermin or rGST. Left panel: Immunofluorescent detection of MYC and Click-iT EdU assay performed on liver sections at day 4 post-injection. Upper right panel: Size of the MYC-positive foci. Median value is indicated with a green line. Number of detected foci on one liversection is shown for each mouse. Lower right panel: Distribution of EdU-positive cells (0–100%) in the foci analyzed. Data information: Scale bar = 100 μm. Data are presented as mean ± SD, all replicates shown are biological replicates. Mann–Whitney test statistical significance is indicated (B, C). Statistical significance of differences in % of EdU-positive foci (C, lower panel) was evaluated using the Kolmogorov–Smirnov test. ns (non significant) $p > 0.05$, **$p < 0.01$, ***$p < 0.001$, ****$p < 0.0001$ DsBc disulfide bond C, rGST recombinant glutathione-S-transferase. Source data are available online for this figure.

Hydrodynamic injections were performed in 6–8-week-old C57BL/6J female mice (Janvier Labs), as described previously (Zhang et al, 1999; Liu et al, 1999). Briefly, 0.1 mL/g of a solution of sterile saline (0.9% NaCl) containing plasmids of interest were injected into lateral tail vein in 8–10 s. In all, 12.5 μg of LentiCRISPRv2-sg*Trp53*, pSBbi-RN-FGF19, pT3-EF1a-MYC, pSBBi-BB-CCND1, or pSBBi-RN-CTNNB1^(S33Y) were injected together with sleeping beauty transposase SB100X (ratio of 5:1). Livers were harvested when mice exhibited signs of tumor development or at a predetermined endpoint ranging from 4 to 52 weeks after injection.

For recombinant protein oncogenicity assay, hydrodynamic injections with pT3-EF1a-MYC and pSBbi-RN together with the plasmid encoding sleeping beauty transposase SB100X (ratio 5:1) were performed. Mice were randomly attributed to one of the three experimental group and injected intra-peritoneally with 1.2 μg of recombinant rAldafermin, rFGF19 or rGST in PBS 0.1% BSA every 12 h, for 4 days. As 3–6 mg of Aldafermin are classically given to patients (60–130 kg), this corresponds to doses ranging from 8 to 100 μg/kg, therefore 0.12–20 μg/mouse. Two hours before mouse euthanasia, 200 μL of 3 mM EdU (Click-iT AF488; ThermoFisher Scientific) in PBS were injected intra-peritoneally.

### Plasmids

All plasmid sequences were validated by whole-plasmid sequencing (SNPsaurus). pCMV(CAT)T7-SB100 was a gift from Zsuzsanna Izsvak (Addgene plasmid #34879). SgTp53 sequence (from Zhang Lab database): 5'-ATAAGCCTGAAAATGTCTCC-3', was cloned into LentiCRISPRv2 vector. Plasmids constructs pSBbi-RN-FGF19 and pSBbi-BB-FGF15 were respectively generated by cloning the human *FGF19* amplified from Huh7 cDNA and the mice *Fgf15* amplified from ileum cDNA onto the pSBbi-RN (Addgene #60519) and pSBbi-BB (Addgene #60521) plasmids digested by the SfiI restriction enzyme. Murine CCND1 and human CTNNB1^(S33Y) open reading frames were cloned into pSBbi-BB and pSBbi-RN, respectively, using the same strategy.

Aldafermin (NGM282) coding sequence was generated from pSBbi-RN-FGF19 plasmid using site directed mutagenesis (Q5® Site-Directed Mutagenesis—New England BioLabs) using the primers FOR 5'-ATCGGGCCTCTGAGGCCATGCGCGACTCGTCGCCCC TCGTGCACTACGGCTGG-3' and REV 5'-CGATGGCCTGACAGG CCTTACTTCTCAAAGCTGGGACTCC-3'.

The plasmids for the bacterial expression of FGF19 and Aldafermin were generated by producing a sequence composed of the T5 promoter followed by the sequence coding for the disulfide bond isomerase (ΔssDsbC), then a T7 promoter followed by a codon-optimized variant of either FGF19 or Aldafermin CDS based

on published work by Choi et al (2023). Those sequences were subsequently cloned in the pQLinkG2 (Addgene #13671) using the XhoI/PasI restriction sites.

### Purification of recombinant proteins

Bacterial expression plasmids for rFGF19/rAldafermin/rGST were transformed into Rosetta-gami™ 2 (DE3) competent cells (Novagen). Bacteria were cultured in LB medium + 100 μg/mL ampicillin and induced with 0.2 mM IPTG at 30 °C for 3 h.

Recombinant FGF19 and Aldafermin proteins were purified without tags using the method described by Choi et al (2023, 2020). To summarize, dried pellets of induced bacteria were solubilized in 30 mL of buffer A1 (20 mM Tris-HCL pH8.0, 1 mM DTT) and sonicated. Lysate was cleared by centrifugation and charged on a 5 mL HiTrap Q HP column (Cytiva) and eluted using a continuous gradient of buffer B1 (20 mM Tris-HCL pH8.0, 1 mM DTT, 1 M NaCl). Fractions containing the protein of interest were pooled, diluted 1:3 with buffer A2 (20 mM Sodium phosphate pH7.0, 150 mM NaCl pH7.0, 1 mM imidazole) and charged on a 1 mL HisTrap (Nickel) column. Protein was eluted with 80 mM imidazole using a step-wise elution with buffer B2 (20 mM Sodium phosphate pH7.0, 150 mM NaCl pH7.0, 100 mM imidazole). Fractions were pooled, diluted 1:6 with buffer A3 (20 mM sodium phosphate pH6.5) and loaded on a 1 mL HiTrap Heparin HP affinity column (Cytiva). Protein was eluted at a 600 mM NaCl using stepwise gradient of buffer B3 (Buffer B: 20 mM sodium phosphate, 1 M NaCl, pH6.5). Recombinant GST was purified using Glutathione Sepharose 4B beads (Cytiva). Bacteria pellet was solubilized in 30 mL of lysis buffer (50 mM Tris-HCl pH7.5, 50 mM NaCl, 0.5 mM DTT) and sonicated. 1 mL of beads were added and incubated at 4 °C for 4 h. Washing steps were done with 30 mL of washing buffer (50 mM Tris-HCl pH7.5, 150 mM NaCl, 0.1% Triton X-100) for 15 min, then final washes were done with a detergent-free buffer (50 mM Tris-HCl pH7.5, 150 mM NaCl, 0.5 mM DTT). Elution was done by with elution buffer (50 mM Tris-HCl pH7.5, 150 mM NaCl, 0.5 mM DTT + 20mM L-glutathion reduced) by 500 μL fractions. Protein purity was assayed using SDS-PAGE and Coomassie blue staining, protein quantification was assayed using human FGF19 ELISA kit (Biovendor, RD191107200R). Samples were diluted 1:2 in PBS + 0.2% BSA and stored at −20 °C.

### Histology assays, image acquisition, and analysis

Livers were fixed for 24 h in 10% neutral buffered formalin, dehydrated, embedded in paraffin and cut into 3 μm-thick sections. Tissue sections were stained with hematoxylin &eosin & saffron

(HES) with Leica autostainer for preliminary analysis. Tumor differentiation and characteristics were reviewed by an expert pathologist (BR). Slides were digitally processed using the Nanozoomer scanner (Hamamatsu).

Immunofluorescence assays were performed after deparaffinization and antigen retrieval in citrate buffer pH6.0 (Sigma Aldrich). The Click-iT AF488 (ThermoFisher Scientific) EdU detection reaction was performed according to the manufacturer's instructions. Labeling of c-Myc-positive cells was performed with human c-Myc antibody (Abcam rabbit ab#32072 dilution 1:200) and anti-rabbit secondary antibody (Thermofisher anti-rabbit IgG AF633, dilution 1:1000). Images were acquired using the Thunder Imager Tissue (Leica) microscope. Quantitative image analysis was performed using the QuPath v0.4.4 software (Bankhead et al, 2017). Foci were hand drawn and positive cell detection were performed using nuclear staining as basis of cell recognition.

### RNA isolation, qPCR, and RNA-Seq analysis

The RNA was extracted from liver tissue and purified using RNeasy mini kit (Qiagen) according to the manufacturer's protocol. Reverse transcription of total RNA (1 μg) was done with QuantiTect Reverse Transcription kit (Qiagen), and cDNA quantified using LC Fast start DNA Master SYBR Green I Mix (Roche) with primers detailed below on LightCycler480 apparatus (Roche). Gene expression levels were normalized with hypoxanthine phospho-ribosyltransferase (*Hprt*). Primer pairs used for qPCR:

Hprt_For 5'-GCAGTACAGCCCCAAAATGG-3'
Hprt_Rev 5'-GGTCCTTTTCACCAGCAAGCT-3'
FGF19_For 5'-CCAGATGGCTACAATGTGTACC-3'
FGF19_Rev 5'-CAGCATGGGCAGGAAATGA-3'
Fgf15_For 5'-TGTCAGATGAAGATCCACTCTTTCTCTA-3'
Fgf15_Rev 5'-GGATTCGGAGGAAGCAGTTG-3'
Afp_For 5'- CTGTCTCAGTCATTCTAAGAATTGCT-3'
Afp_Rev 5'- CTCCTCGATGTGTTTCTGC-3'
Gpc3_For 5'-AGGACTGTGGCCGTATG-3'
Gpc3_Rev 5'-GCAATAACCACCGCAAGG-3'
Cyp7a1_For 5'-CTGCAACCTTCTGGAGCTTA-3'
Cyp7a1_Rev 5'- ATCTAGTACTGGCAGGTTGTTT-3'

For RNA-Seq analysis, RNA was extracted from snap frozen individual tumors, using RNeasy mini kit (Qiagen) with DNAse treatment. RNA integrity was validated using RNA BioAnalyzer (Agilent), all RIN > 9.0. The preparation of the library was done with the TruSeq Stranded mRNA Sample Preparation kit (Illumina). The sequencing was performed in an Illumina Hiseq 2500 sequencer by the Sequencing Platform of Montpellier (GenomiX, MGX, France), with 150 base pairs (bp) paired end reads to an estimated depth of 70 million reads per sample. FastQC was used for quality control on raw sequence data. All the reads that passed the quality control were aligned to the mouse reference genome (Mus musculus NCBI Mm39) with HiSAT2 and the counts per gene were quantified using the tool htseq-count. The NCBI RefSeq mouse genome annotations (accessed 27/02/2023) were used for establishing the coordinates of each gene and their corresponding transcripts. Differential gene expression analysis was performed in R using the EdgeR package (Robinson et al, 2010).

Transcripts with less than one count per million (CPM) were discarded from the analysis. Fitting the globalized linear model

(GLM) was done using the tagwise dispersion. Transcript with at least two comparisons under the adjusted p-value threshold of 0.05 were considered as differentially expressed genes (DEG). Hierarchical clustering was applied to the samples and to the transcripts contributing to the dimension 2 of the PCA (contribution >0.003 and |correlation| > 0.5) using the 1-Pearson correlation coefficient as distance and Ward's criterion for agglomeration. The clustering results are illustrated as a heatmap of normalized counts per millions, centered and scaled. Pathways enriched in FGF19/Aldafermin driven tumors were determined by GSEA analysis: Aldafermin + MYC and FGF19 + MYC tumors data were pooled and compared to sgTrp53+MYC and Liver samples. Pathways of size between 21 and 1000 genes were considered, with a threshold of *p* value <0.01. To minimize contribution of p53 deficiency, we focus our subsequent analyses on the 18 GSEA pathways enriched specifically in Aldafermin/FGF19 + MYC vs sgTrp53 + MYC tumors.

### Protein modeling and alignment

Protein modeling of FGF19 and Aldafermin were done using AlphaFold predictions via the ChimeraX software (Pettersen et al, 2021). Protein 3D alignment was made using the RCSB PDB Pairwise Structure Alignment website (https://www.rcsb.org/alignment, last accessed 20 July 2023) using the jFATCAT method (Ye and Godzik, 2003).

### Experimental design and statistical analysis

The investigators were blinded every time it was possible. No specific exclusion criteria were applied to the data analysis, all collected data points were included in the analyses. Data sets were tested with two-tailed unpaired Student *t* tests or Mann–Whitney *U*-tests, correlations were analyzed with Pearson's $\chi^2$ test using Prism Software version 8 (GraphPad). Significant *p* values are shown as: *$p < 0.05$, **$p < 0.01$, ***$p < 0.001$, and ****$p < 0.0001$. Calculation of sample size for animal experiments was performed with G*Power software, with a detection power of 0.8, and $\alpha < 0.05$. FEWER *p* values and associated FDR *q*-values were calculated using R « qvalue » package.

Data visualization and statistical analysis were performed using R software version 3.5.1 (R Foundation for Statistical Computing, Vienna, Austria. https://www.R-project.org) and Bioconductor packages. GSEA analysis were performed with ViSEAGO package. Comparisons of the mRNA expression levels between groups were assessed using Mann–Whitney *U*-test. Spearman's rank-order correlation was used to test the association between continuous variables. Univariate survival analysis was performed using Kaplan–Meier curve with log-rank test. Kolmogorov–Smirnov test was used to compare the distribution of experimental groups.

# Data availability

The RNA-sequencing data have been deposited in the Gene Expression Omnibus (GEO, NCBI) repository and are accessible through GEO Series accession number GSE242953.

## The paper explained

### Problem

Fibroblast growth factor 19 (FGF19) is a hormone with well-documented beneficial metabolic effects. However, it is also known to promote carcinogenesis in hepatocellular carcinoma (HCC), the main primary liver tumor. FGF19 analogs (such as Aldafermin), reported to be devoid of oncogenic properties, have been developed and are currently being tested in patients with metabolic dysfunction-associated steatohepatitis (MASH). However, the potential synergy between FGF19, or its analogs, and other frequent oncogenic events in liver carcinogenesis has so far been understudied.

### Results

We used preclinical animal models of liver carcinogenesis to test possible oncogenic cooperation between FGF19 and Aldafermin with oncogenic events frequently encountered in human HCC. We identified a strong cooperation of both FGF19 and Aldafermin with an increased expression of MYC, an oncogene activated in many, if not all, malignancies. The tumors developed by the animals shared molecular and histological characteristics with human HCC, which arises mostly in chronically injured liver, such as in patients suffering from MASH.

### Impact

Overall, our preclinical results strongly argue for caution in the clinical use of FGF19 analogs for treatment of metabolic liver dysfunction. This is especially true for patients suffering from chronic liver diseases, who are at risk of harboring genetic alterations that might interact with FGF19 analogs in the process of liver tumorigenesis.

# Peer review information

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

## Acknowledgements

We acknowledge Montpellier Biocampus facilities: the "Réseau d'Histologie Expérimentale de Montpellier" - RHEM facility supported by SIRIC Montpellier Cancer Grant INCa_Inserm_DGOS_12553, the European regional development foundation and the Occitanian region (FEDER-FSE 2014-2020 Languedoc Roussillon) for processing some of our animal tissues; the MGX facility for the RNAseq analysis, and the MRI imaging platform. We are grateful to Olivier Coux and Aymeric Bailly for their help in producing rFGF19 and rAldafermin recombinant proteins and to Geun-Joong Kim's Lab. We thank Georges-Philippe Pageaux for helpful discussions and Michael Hahne for support. This work was funded by ITMO Cancer (EVA-Plan cancer INSERM

HTE201610), Association Française pour l'Etude du Foie (AFEF) AAP_2021 and supported by SIRIC Montpellier Cancer Grant INCa_Inserm_DGOS_12553, la Fondation ARC and INCa (2019-134). JU-B was funded by SIRIC Montpellier. The funders had no role in study design, data collection and analysis or publication process.

## Author contributions

**José Ursic-Bedoya**: Conceptualization; Investigation; Writing—original draft; Writing—review and editing. **Guillaume Desandre**: Conceptualization; Investigation; Visualization; Writing—review and editing. **Carine Chavey**: Investigation; Writing—review and editing. **Pauline Marie**: Formal analysis. **Arnaud Polizzi**: Formal analysis. **Benjamin Rivière**: Anatomopathological analyses. **Herve Guillou**: Formal analysis. **Eric Assénat**: Conceptualization. **Urszula Hibner**: Conceptualization; Funding acquisition; Writing—original draft; Writing—review and editing. **Damien Gregoire**: Conceptualization; Supervision; Funding acquisition; Visualization; Writing—original draft; Project administration; Writing—review and editing.

## Disclosure and competing interests statement

The authors declare no competing interests.

# Expanded View Figures

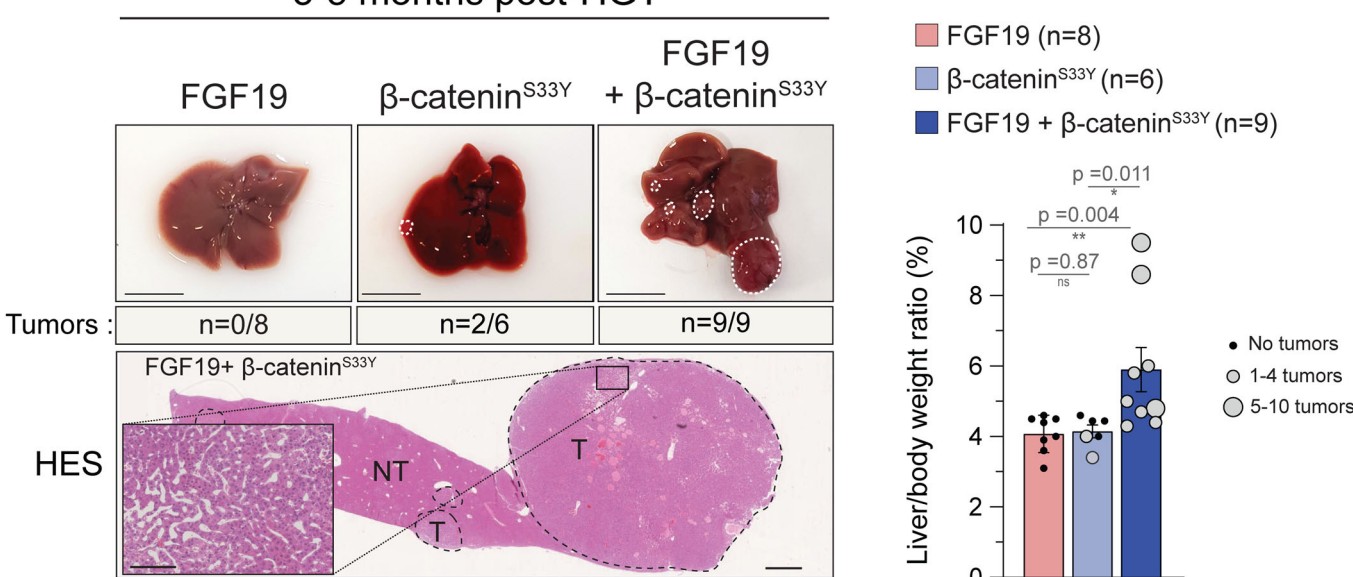

**Figure EV1. FGF19 cooperates with β-catenin[S33Y] to induce hepatic carcinogenesis.**

Representative livers and HES-stained sections from mice following hydrodynamic gene transfer with either FGF19, β-catenin[S33Y] or both. Tumor incidence is indicated. Liver/body weight ratio for each mouse, with dot sizes proportional to tumor burden. Scale bar for zoom: 200 μm. Scale bar for large view: 1 cm. Mann–Whitney test statistical significance is indicated. Data are represented as mean ± SD. ns (non significant) $p > 0.05$, $*p < 0.05$, $**p < 0.01$.

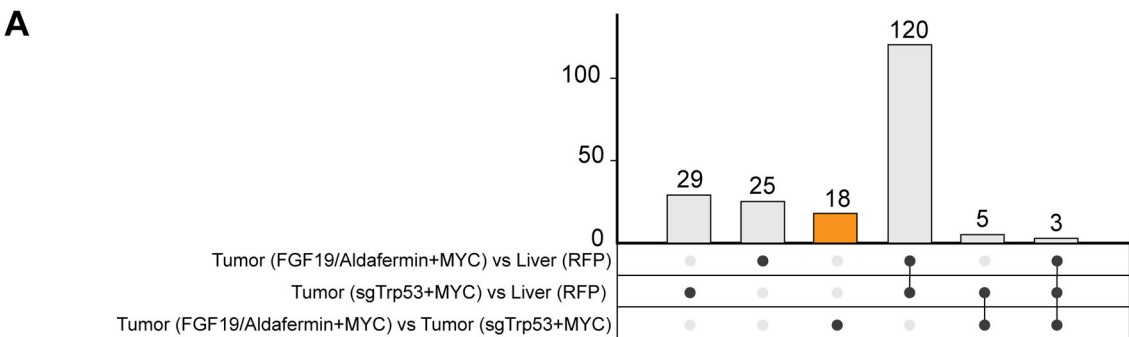

**Figure EV2. GSEA Pathways enriched in FGF19/Aldafermin + MYC driven tumors.**

(**A**) Upset plot depicting the number of unique and shared GSEA pathways enriched in each sample groups analyzed by RNAseq. FGF19 + MYC and Aldafermin + MYC samples were pooled for this analysis. (**B**) GSEA analysis of the 18 pathways specifically enriched in FGF19/Aldafermin + MYC vs sgTrp53 + MYC tumors. Kolmogorov–Smirnov statistical test significance is indicated. Data information: Threshold of significance for GSEA: $p < 0.01$. Complementary data are provided in Dataset EV1.

