## [Peer Review File · EMBO Molecular Medicine]

FGF19 and its analog Aldafermin cooperate with MYC to induce aggressive hepatocarcinogenesis

José Ursic-Bedoya, Guillaume Desandre, Carine Chavey, Pauline Marie, Arnaud Polizzi, Benjamin Rivière, Herve Guillou, Eric Assénat, Urszula Hibner, and Damien Gregoire

DOI: [10.15252/emmm.202318694](https://doi.org/10.15252/emmm.202318694)

Corresponding author: Damien Gregoire (damien.gregoire@igmm.cnrs.fr)

Review Timeline:

Submission Date:	15th Sep 23
Editorial Decision:	9th Oct 23
Revision Received:	8th Dec 23
Editorial Decision:	18th Dec 23
Revision Received:	18th Dec 23
Accepted:	20th Dec 23

Editor: Lise Roth

Transaction Report:

9th Oct 2023

Dear Dr. Gregoire,

Thank you for the submission of your manuscript to EMBO Molecular Medicine. We have now received feedback from the three reviewers who agreed to evaluate your manuscript. As you will see from the reports below, the referees acknowledge the interest of the study and are overall supporting publication of your work pending appropriate revisions.

Addressing the reviewers' concerns in full will be necessary for further considering the manuscript in our journal, and acceptance of the manuscript will entail a second round of review. In particular, the referees all agreed that based on the RNAseq data, further mechanistic insight should be provided.

EMBO Molecular Medicine encourages a single round of revision only and therefore, acceptance or rejection of the manuscript will depend on the completeness of your responses included in the next, final version of the manuscript. For this reason, and to save you from any frustrations in the end, I would strongly advise against returning an incomplete revision.

We are expecting your revised manuscript within three months, if you anticipate any delay, please contact us.

We require:

4) A .docx formatted letter INCLUDING the reviewers' reports and your detailed point-by-point responses to their comments. As part of the EMBO Press transparent editorial process, the point-by-point response is part of the Review Process File (RPF), which will be published alongside your paper.

5) A complete author checklist, which you can download from our author guidelines (<https://www.embopress.org/page/journal/17574684/authorguide#submissionofrevisions>). Please insert information in the checklist that is also reflected in the manuscript. The completed author checklist will also be part of the RPF.

6) For data quantification: please specify the name of the statistical test used to generate error bars and P values, the number (n) of independent experiments (specify technical or biological replicates) underlying each data point and the test used to calculate p-values in each figure legend. The figure legends should contain a basic description of n, P and the test applied. Graphs must include a description of the bars and the error bars (s.d., s.e.m.). Please provide exact p values.

7) Our journal encourages inclusion of *data citations in the reference list* to directly cite datasets that were re-used and obtained from public databases. Data citations in the article text are distinct from normal bibliographical citations and should directly link to the database records from which the data can be accessed. In the main text, data citations are formatted as follows: "Data ref: Smith et al, 2001" or "Data ref: NCBI Sequence Read Archive PRJNA342805, 2017". In the Reference list, data citations must be labeled with "[DATASET]". A data reference must provide the database name, accession number/identifiers and a resolvable link to the landing page from which the data can be accessed at the end of the reference. Further instructions are available at .

8) We replaced Supplementary Information with Expanded View (EV) Figures and Tables that are collapsible/expandable online. A maximum of 5 EV Figures can be typeset. EV Figures should be cited as 'Figure EV1, Figure EV2' etc... in the text and their respective legends should be included in the main text after the legends of regular figures.

- For the figures that you do NOT wish to display as Expanded View figures, they should be bundled together with their legends in a single PDF file called *Appendix*, which should start with a short Table of Content. Appendix figures should be referred to in

the main text as: "Appendix Figure S1, Appendix Figure S2" etc.

9) The paper explained: EMBO Molecular Medicine articles are accompanied by a summary of the articles to emphasize the major findings in the paper and their medical implications for the non-specialist reader. Please provide a draft summary of your article highlighting

10) For more information: There is space at the end of each article to list relevant web links for further consultation by our readers. Could you identify some relevant ones and provide such information as well? Some examples are patient associations, relevant databases, OMIM/proteins/genes links, author's websites, etc...

11) Author contributions: CRediT has replaced the traditional author contributions section because it offers a systematic machine readable author contributions format that allows for more effective research assessment. Please remove the Authors Contributions from the manuscript and use the free text boxes beneath each contributing author's name in our system to add specific details on the author's contribution. More information is available in our guide to authors.

12) Disclosure statement and competing interests: We updated our journal's competing interests policy in January 2022 and request authors to consider both actual and perceived competing interests. Please review the policy <https://www.embopress.org/competing-interests> and update your competing interests if necessary.

13) Every published paper now includes a 'Synopsis' to further enhance discoverability. Synopses are displayed on the journal webpage and are freely accessible to all readers. They include a short stand first (maximum of 300 characters, including space) as well as 2-5 one-sentences bullet points that summarizes the paper. Please write the bullet points to summarize the key NEW findings. They should be designed to be complementary to the abstract - i.e. not repeat the same text. We encourage inclusion of key acronyms and quantitative information (maximum of 30 words / bullet point). Please use the passive voice. Please attach these in a separate file or send them by email, we will incorporate them accordingly.

14) As part of the EMBO Publications transparent editorial process initiative (see our Editorial at <http://embomolmed.embopress.org/content/2/9/329>), EMBO Molecular Medicine will publish online a Review Process File (RPF) to accompany accepted manuscripts.

In the event of acceptance, this file will be published in conjunction with your paper and will include the anonymous referee reports, your point-by-point response and all pertinent correspondence relating to the manuscript. Let us know whether you agree with the publication of the RPF and as here, if you want to remove or not any figures from it prior to publication. Please note that the Authors checklist will be published at the end of the RPF.

I look forward to receiving your revised manuscript.

Yours sincerely,

Lise Roth

**** Reviewer's comments ****

Referee #1 (Remarks for Author):

In this work, Ursic-Bedoya and collaborators further evaluate the hepatic pro-tumorigenic activity of FGF19 in mouse models that combine additional oncogenic effectors, such as CCND1 or MYC overexpression, or the expression of mutant CTNNB1. Consistent with previous observations in which FGF19 has been overexpressed in mice with ongoing liver injury, fibrosis, and inflammation, such as Mdr2 KO mice, FGF19 significantly enhanced the tumorigenicity of MYC or mutant CTNNB1. Interestingly, this effect is now extended to FGF15, the murine orthologue of FGF19. Nevertheless, the most relevant finding of this work is the fact that the FGF19 analog Aldafermin, an FGF19 variant that in principle is devoid of pro-tumorigenic activities, also behaves as a potent oncogenic factor, synergizing with MYC overexpression. Aldafermin (also known as NGM282) was previously tested in mouse models, including Mdr2 KO mice, and no oncogenic effects were observed. Aldafermin was designed to treat different types of chronic liver diseases and metabolic disorders (NASH, NASH-cirrhosis, PSC, PBC), and is/has been tested in different clinical trials. Therefore, the current findings by Ursic-Bedoya and colleagues are relevant from a clinical point of view. Overall, the data presented are clear and convincing, and the manuscript is well written. Nevertheless, there are some issues for the authors to consider which, if properly addressed can enhance the significance of this worthy report.

Major points

1. The authors should revise the quality of Fig. 1. At least in the PDF copy I received for review, there is considerable overlapping in the labels and figures of the different panels, mainly panels A, C, and E, which makes it very difficult to understand the results shown.
2. Regarding Fig. 1D, the expression of FGF19 and FGF15 in the different mouse liver samples tested should be analyzed at least by qRT-PCR.
3. Regarding data in Fig. 2B, the oncogenicity of Aldafermin, MYC, and Aldafermin + MYC HGT should be tested, analyzed, and shown side-by-side in the same experiment.
4. In Fig. 3A, it is not indicated what samples are analyzed in the Coomassie-Blue stained gel shown.
5. Regarding Fig. 3C, the design of the experiment is not well described in the text (neither in the Methods section nor in the Results section, let alone the figure legends, which are very poor in general). Was the MYC plasmid co-injected with the Sleeping Beauty plasmid? How was the administration of EdU performed?
6. In the experiment shown in Fig. 3C, the authors should include a control group of mice treated with an "irrelevant" protein that has been also expressed and purified from bacteria, at equivalent doses as those of FGF19 and Aldafermin.
7. Given the relevance of the findings, the pro-carcinogenic mechanisms of Aldafermin in combination with MYC should be better elucidated.
8. In support of their current observations, the authors may consider discussing a previous study that already addressed the role of endogenous FGF15/19 in mice undergoing experimental carcinogenesis associated with chronic injury and inflammation (PMID: 25346390).

Minor point

- The reference BouSaba et al 2023 is not included in the reference list.

Referee #2 (Comments on Novelty/Model System for Author):

Study design: one the most common mutation in HCC concerns TERT (found in ~60% of cases). Why was not this selected in the study, regarding the HCC mouse model? What about PTEN? I suggest that a solid rationale for the choice of the suitable animal models should start from the frequency of the relative mutation in human HCC. Otherwise, the authors should better justify their choice of mice models.

Referee #2 (Remarks for Author):

This is an interesting and potentially important study.

I have several concerns to be addressed, in no particular order:

- The statistical soundness of the survival curves in Figure 1C is unclear. Group size is small, and p values are not indicated.
- As a more general consideration, there is no calculation of sample size in most animal experiments; a power calculation to assess the adequacy of the sample size has not been described. This is possible and indispensable to do, as the frequency of tumor development in the models used has been described. Results may be reevaluated on the basis of the correct sample size.
- The RNA-Seq dataset in Figure 2C is used only to describe the overlap between Aldafermin+MYC with FGF19+MYC. RNA-Seq can offer many other insights. I suggest to use the data for the potential they have and to offer more data and insights on the tumor transcriptomics.
- Study design: one the most common mutation in HCC concerns TERT. Why was not this selected in the study? What about PTEN?
- Page 7: it is mentioned that you sought to reproduce the human administration of aldafermin in mice. Is i.p. injection of several ug of aldafermin what would be administered in humans? You should show a dose/weight species parallelism, if a comparison is to be made.
- Can you comment better in the Introduction what were the mixed results of the Aldafermin trials?
- The reference Tao Z et al is incomplete.
- The reference BouSaba et al is not there.

Referee #3 (Comments on Novelty/Model System for Author):

I have no suggestions

Referee #3 (Remarks for Author):

The manuscript titled " FGF19 and its analog Aldafermin cooperate with MYC to induce aggressive hepatocarcinogenesis" aims to analyze the effects of the hormone FGF19/15 and its analog aldafermin on tumor-promoting activity in a model of hepatocarcinoma induced by various oncogenic alterations such as p53 inactivation, CTNNB1 mutation, CCND1, or MYC overexpression. The results were remarkable and aligned with the manuscript's proposed title. Nevertheless, some shortcomings were identified, and the work could be enhanced by addressing the following minor points:

1. Figure 1 is distorted in the revised manuscript.
2. In the first section of the abstract, the word "notably" is repeated twice.
3. The inclusion of the comparison Aldafermin+MYC vs FGF19+MYC in the Euler diagram (Fig2C) is confusing. If it is to be retained, it is recommended to explain the numbers of genes 36, 40, and 95.
4. Standardize the use of HCC or hepatocellular carcinoma.

- **Referee #1 (Remarks for Author):**

In this work, Ursic-Bedoya and collaborators further evaluate the hepatic pro-tumorigenic activity of FGF19 in mouse models that combine additional oncogenic effectors, such as CCND1 or MYC overexpression, or the expression of mutant CTNNB1. Consistent with previous observations in which FGF19 has been overexpressed in mice with ongoing liver injury, fibrosis, and inflammation, such as Mdr2 KO mice, FGF19 significantly enhanced the tumorigenicity of MYC or mutant CTNNB1. Interestingly, this effect is now extended to FGF15, the murine orthologue of FGF19. Nevertheless, the most relevant finding of this work is the fact that the FGF19 analog Aldafermin, an FGF19 variant that in principle is devoid of pro-tumorigenic activities, also behaves as a potent oncogenic factor, synergizing with MYC overexpression. Aldafermin (also known as NGM282) was previously tested in mouse models, including Mdr2 KO mice, and no oncogenic effects were observed. Aldafermin was designed to treat different types of chronic liver diseases and metabolic disorders (NASH, NASH-cirrhosis, PSC, PBC), and is/has been tested in different clinical trials. Therefore, the current findings by Ursic-Bedoya and colleagues are relevant from a clinical point of view. Overall, the data presented are clear and convincing, and the manuscript is well written. Nevertheless, there are some issues for the authors to consider which, if properly addressed can enhance the significance of this worthy report.

We thank the reviewer for his/her positive comments and especially for the appreciation of the clinical relevance of our work.

Major points:

1. The authors should revise the quality of Fig. 1. At least in the PDF copy I received for review, there is considerable overlapping in the labels and figures of the different panels, mainly panels A, C, and E, which makes it very difficult to understand the results shown.

We apologize for this technical issue that occurred during the PDF generation process on the journal submission website. This is rather frustrating, considering the significant amount of time we invested in producing high-quality figures. We are grateful to the reviewer that it did not negatively impact the evaluation of our work.

2. Regarding Fig. 1D, the expression of FGF19 and FGF15 in the different mouse liver samples tested should be analyzed at least by qRT-PCR.

This is an important control that we have in fact performed and should have included in the initial version of the manuscript. These data are now included in the new version of Figure 1F.

3. Regarding data in Fig. 2B, the oncogenicity of Aldafermin, MYC, and Aldafermin + MYC HGT should be tested, analyzed, and shown side-by-side in the same experiment.

The oncogenicity of the different combinations was in fact tested and analyzed within the same experiments. We used between 2 or 3 plasmid preparations and 8-13 animals in three

independent experiments. The reason the RFP+MYC HGT results were not shown in the original Fig. 2B is that they were already presented in Figure 1D-E, and we were hesitant to show them again in Figure 2B. This is now corrected, and RFP+MYC are now shown side-by-side with Aldafermin, and Aldafermin+MYC HGT in the new version of Figure 2B.

4. In Fig. 3A, it is not indicated what samples are analyzed in the Coomassie-Blue stained gel shown.

We apologize for this lack of clarity. The rAldafermin Coomassie-Blue stained gel was included for the sole purpose of illustration of the recombinant protein purification approach. The samples analyzed and fractions are now indicated, and the FPLC profiles and Coomassie-Blue stained gels for rFGF19 and rAldafermin purifications are presented in the Raw data section.

5. Regarding Fig. 3C, the design of the experiment is not well described in the text (neither in the Methods section nor in the Results section, let alone the figure legends, which are very poor in general). Was the MYC plasmid co-injected with the Sleeping Beauty plasmid? How was the administration of EdU performed?

We acknowledge, and regret, that the description of the experiment with recombinant proteins was not sufficiently detailed. The MYC plasmid was co-injected with SB100 plasmid, at similar doses than in experiments of Figures 1 and 2. EdU was injected intra-peritoneally two hours before liver collection, following manufacturer's instructions and classical protocols. We have changed the text to specify these points, both in methods and results sections. In addition, we improved the figure legends, within the restrictions of the EMBO press format.

6. In the experiment shown in Fig. 3C, the authors should include a control group of mice treated with an "irrelevant" protein that has been also expressed and purified from bacteria, at equivalent doses as those of FGF19 and Aldafermin.

In the revised version of the manuscript, we present data obtained in an experiment we have now performed. A control group of mice was treated with 1.2 µg of GST (26 kDa, same size as FGF19 and Aldafermin), produced via the same plasmid backbone and in the same strain of bacteria. The injection of this control "irrelevant" recombinant protein did not generate MYC-positive foci, similarly to what was observed with previous control group injected with PBS+BSA. The results of this important experiment are now shown in Figure 3C.

7. Given the relevance of the findings, the pro-carcinogenic mechanisms of Aldafermin in combination with MYC should be better elucidated.

We fully agree that pro-carcinogenic mechanisms of FGF19 and Aldafermin in combination with MYC overexpression need to be further explored. To provide initial insights into this matter, we have now compared transcriptomes of FGF19/Aldafermin-driven tumors with those generated by HGT of sgTrp53+MYC. We opted for the use of tumors with inactivated p53 tumor suppressor since as HGT with MYC alone rarely gave rise to tumors in our experiments. We were careful to eliminate, as much as possible, the effects due to p53 deficiency to look for pathways enriched specifically by the hormone and its analog (See Upset plot now provided as EV3A). This led to the identification of 13 GSEA pathways (Figure 1F, EV3B). Among them, induction of EMT, SMAD signaling, vasculogenesis and extracellular matrix remodeling are particularly interesting, since they are all plausible

candidates for pro-carcinogenic mechanisms of FGF19/Aldafermin. Further investigations, beyond the scope of this report, will however be necessary to more fully elucidate the molecular mechanisms involved in the oncogenic cooperation with MYC.

8. In support of their current observations, the authors may consider discussing a previous study that already addressed the role of endogenous FGF15/19 in mice undergoing experimental carcinogenesis associated with chronic injury and inflammation (PMID: 25346390).

We apologize for this omission, this reference has now been included and we have modified the text in accordance (Results section, page 6, line 4)

Minor point:

- The reference BouSaba et al 2023 is not included in the reference list.

We apologize for this error that has now been corrected.

Referee #2 (Comments on Novelty/Model System for Author):

Study design: one the most common mutation in HCC concerns TERT (found in ~60% of cases). Why was not this selected in the study, regarding the HCC mouse model? What about PTEN? I suggest that a solid rationale for the choice of the suitable animal models should start from the frequency of the relative mutation in human HCC. Otherwise, the authors should better justify their choice of mice models.

The referee is of course right: TERT promoter mutation, or TERT amplification, are indeed very common in HCC. We have not included them our study since TERT dependency is less crucial in mouse than in human. Fundamental differences in telomere damage signaling have notably been reported, that bears on the use of mouse models for the telomere tumor suppressor pathway (Smogorzewska et al., 2002, PMID: 12169636). Accordingly, hydrodynamic injections with TERT overexpression (combined with MYC or CTNNB1) had little effect on tumorigenesis (Molina Sanchez et al., 2020, PMID32814112).

We agree that PTEN deletion would be interesting to test in combination with FGF19 overexpression. It is considerably less frequent in human HCC compared to MYC or beta-catenin activation or p53 inactivation (3-6%) and was therefore not considered a priority within our rationale based on frequency of oncogenic events in human HCC. Indeed, in this study, apart from the TERT mutations from the reasons explained above, we chose to investigate the most frequently affected genes in HCC patients: Tp53, CTNNB1 and MYC (Zucman Rossi et al., 2015; Molina-Sanchez et al. 2020). We added the CCDN1 overexpression in our study design due to its genomic co-amplifications with FGF19 (Sawey et al. 2011). A sentence in the first paragraph of the manuscript has been added to clarify the rationale for the choice of oncogenic events.

- Referee #2 (Remarks for Author):

This is an interesting and potentially important study.

I have several concerns to be addressed, in no particular order:

- The statistical soundness of the survival curves in Figure 1C is unclear. Group size is small, and p values are not indicated.

We apologize for this lack of precision. We have now included the value of the log-rank test for the Kaplan-Meier curve in the Figure 1C (p-value <0.0001). The group sizes are between n=5 and n=12, respecting the group size calculations and following ethical considerations of reducing the number of animals used (3R).

-As a more general consideration, there is no calculation of sample size in most animal experiments; a power calculation to assess the adequacy of the sample size has not been described. This is possible and indispensable to do, as the frequency of tumor development in the models used has been described. Results may be reevaluated on the basis of the correct sample size.

For the authorization of the project by animal welfare committee, we calculated sample size (using G*Power software) with the classically used detection power of 0.80. For example, for Figure 1D-E, we calculated that to detect a difference of 30% in means between groups with a Wilcoxon-Mann-Whitney test ($\alpha < 0.05$), the minimum number of animals per group should be 6. We now mention calculation of sample size approach in the Methods section.

- The RNA-Seq dataset in Figure 2C is used only to describe the overlap between Aldafermin+MYC with FGF19+MYC. RNA-Seq can offer many other insights. I suggest to use the data for the potential they have and to offer more data and insights on the tumor transcriptomics.

We agree that RNA-Seq data were used at minimum in the previous version of the manuscript. Basically, it served to show that FGF19 and Aldafermin combined with MYC generate tumors with indistinguishable transcriptomes. We have now extended the analysis the RNAseq data by comparing the transcriptomic profiles of FGF19/Aldafermin driven tumors with those generated by HGT with MYC+sgTrp53 (as HGT with MYC alone rarely gave rise to tumors in our experiments) (Figure 1C-E). The results confirmed the close proximity of FGF19+MYC and Aldafermin+MYC transcriptomic profiles. Further insight into biological pathways deregulated by the hormone and its analog were provided by GSEA analysis. We were careful to eliminate, as much as possible, the effects of p53 deficiency (See Upset plot now provided as EV3A). We identified 13 GSEA pathways (Figure 1F, EV3B), among which, induction of EMT, SMAD signaling, vasculogenesis and extracellular matrix remodeling are particularly interesting since they could be involved in pro-carcinogenic mechanisms of FGF19/Aldafermin.

- Study design: one the most common mutation in HCC concerns TERT. Why was not this selected in the study? What about PTEN?

Please see our response above, in the "Comments on Model system" section.

- Page 7: it is mentioned that you sought to reproduce the human administration of aldafermin in mice. Is i.p. injection of several ug of aldafermin what would be administered in humans? You should show a dose/weight species parallelism, if a comparison is to be made.

We agree, this is a complex question. Indeed, in clinical trials with Aldafermin, 1 to 6 mg are delivered subcutaneously *q.d* to subjects with a widely variable patient's weight (60 to 130 kg), which therefore correspond to doses ranging from 8 to 100 µg/kg. We calculated the dose we injected for a mouse weighing 20 g (standard weight in our animal facility for female C57Bl/6 mice at this age). The dose/weight equivalence hence varies between 0.12 µg to 20 µg per mouse, and we settled for values of 2 or 2.4 µg in our experiments, taking into consideration the Animal Equivalence Dose (see for example Nair et al., 2026 PMID:27057123). Considering patient's plasmatic concentrations published in the literature (see Supplementary Figure S5 in Harrison et al. 2022, *Lancet Gastroenterol Hepatol.*, doi: 10.1016/S2468-1253(22)00017-6), we then set up our experimental setting to injections every 12 hours. Interspecies allometric scaling for dose conversion from animal to human studies is still debated, but it is important to note that the concentrations of Aldafermin and FGF19 used in this experiment are in the low range of what is given to patients, making the observed results compelling. As suggested, the dose/weight species parallelism is now mentioned in the Methods section.

- Can you comment better in the Introduction what were the mixed results of the Aldafermin trials?

We recognize the formulation was not clear and have changed the text to describe more precisely the results of clinical trials using Aldafermin (page 4 lines 17-23).

- The reference Tao Z et al. is incomplete.

- The reference BouSaba et al. is not there.

We apologize for these errors, both references have now been corrected.

• Referee #3 (Remarks for Author):

The manuscript titled " FGF19 and its analog Aldafermin cooperate with MYC to induce aggressive hepatocarcinogenesis" aims to analyze the effects of the hormone FGF19/15 and its analog aldafermin on tumor-promoting activity in a model of hepatocarcinoma induced by various oncogenic alterations such as p53 inactivation, CTNNB1 mutation, CCND1, or MYC overexpression. The results were remarkable and aligned with the manuscript's proposed title. Nevertheless, some shortcomings were identified, and the work could be enhanced by addressing the following minor points:

1. Figure 1 is distorted in the revised manuscript.

We apologize for this technical issue that occurred during the PDF generation process on the journal submission website. It is indeed frustrating, considering the significant amount of time we invest in producing high-quality figures. We are grateful to the reviewer that it did not negatively impact the evaluation of our work.

2. In the first section of the abstract, the word "notably" is repeated twice.

Sorry for this lack of elegance in our style, we have now changed the first sentence to avoid the repetition.

3. The inclusion of the comparison Aldafermin+MYC vs FGF19+MYC in the Euler diagram (Fig2C) is confusing. If it is to be retained, it is recommended to explain the numbers of genes 36, 40, and 95.

We acknowledge that the Euler diagram presentation lacked clarity, mainly due to the close proximity of Aldafermin+MYC vs FGF19+MYC transcriptomes. We therefore removed this representation, as PCA and correlation analyses already clearly illustrate that these tumors have very similar transcriptional programs.

4. Standardize the use of HCC or hepatocellular carcinoma.

The term HCC is now used throughout the entire article.

18th Dec 2023

Dear Dr. Gregoire,

Thank you for submitting your revised manuscript. We have now received the reports from the referees who re-reviewed your manuscript, and as you will see below, they are supportive of publication. I will therefore be able to accept your manuscript once the following points will be addressed:

1/ Manuscript text:

- Please correct the spelling mistake mentioned by referee #1.
- Please remove the red text, and only keep in track changes mode any new modification.
- Materials and Methods:
 - o Mice experiments: please indicate the origin and strain of the mice.
 - o Antibodies: please provide dilutions/concentrations.
 - o Statistics: please include a statement on inclusion/exclusion criteria and correct the checklist accordingly.
- Data Availability section: This section should be placed after the Materials and Methods. Please note that the datasets must be public before acceptance of the manuscript. Please add a URL link to the deposited dataset. Please remove "All newly created materials are made available to the community."
- Acknowledgements: Please make sure that the information provided in this section matches the information provided in the submission system (currently, EVA-22 Plan cancer INSERM THE is missing from the submission system).
- Please rename "Conflict of interest" to "Disclosure statement and competing interests": We updated our journal's competing interests policy in January 2022 and request authors to consider both actual and perceived competing interests. Please review the policy <https://www.embopress.org/competing-interests> and update your competing interests if necessary.

2/ Figures:

- A reference is missing for Fig EV1 in the main manuscript text.
- Table EV2 should be renamed "Dataset EV1"; its legend should be removed from the manuscript and added to the csv file.
- Figure legends:
 1. Please indicate the statistical test used for data analysis in the legends of figure 1c; EV 2b.
 2. Please note that in figures 1c, e-f; 2b; 3b-c; there is a mismatch between the annotated p values in the figure legend and the annotated p values in the figure file that should be corrected.
 3. Please note that the error bar is not defined in the legend of figure 2b.

3/ As part of the EMBO Publications transparent editorial process initiative (see our Editorial at <http://embomolmed.embopress.org/content/2/9/329>), EMBO Molecular Medicine will publish online a Review Process File (RPF) to accompany accepted manuscripts.

This file will be published in conjunction with your paper and will include the anonymous referee reports, your point-by-point response and all pertinent correspondence relating to the manuscript. Let us know whether you agree with the publication of the RPF.

I look forward to receiving your revised manuscript.

With kind regards,

Lise

***** Reviewer's comments *****

Referee #1 (Remarks for Author):

No further comments. Nevertheless, please revise the spelling, on page 6, line 7, "FG15" should be FGF15.

Referee #2 (Remarks for Author):

All comments have been satisfactorily addressed

The authors addressed the minor editorial issues.

20th Dec 2023

Dear Dr. Gregoire,

Thank you for submitting the revised files. I am pleased to inform you that your manuscript is accepted for publication and is now being sent to our publisher to be included in the next available issue of EMBO Molecular Medicine!

Congratulations on your interesting work!

With kind regards,

Lise
